# Evidence for a Negative Loss Spiral between Co-Worker Social Support and Burnout: Can Psychosocial Safety Climate Break the Cycle?

**DOI:** 10.3390/healthcare11243168

**Published:** 2023-12-14

**Authors:** Sarven S. McLinton, Stephanie D. Jamieson, Michelle R. Tuckey, Maureen F. Dollard, Mikaela S. Owen

**Affiliations:** Psychosocial Safety Climate: Global Observatory, Centre for Workplace Excellence, Justice and Society, University of South Australia, Magill Campus, P.O. Box 2471, Adelaide, SA 5001, Australia

**Keywords:** psychosocial safety climate, social support, burnout, loss cycles, loss spiral, frontline healthcare, nursing

## Abstract

Previous research suggests that co-worker social support predicts burnout, but this relationship may be far more complex, with the potential for a reciprocal cycle of loss. Leading research on loss spirals has explicitly called for more research on interindividual factors such as social support and, by extension, how interventions that operate on these interpersonal resources could play a role in primary and secondary prevention (i.e., intervening in cycles of loss). In this study, we explore the reciprocal relationship between burnout and co-worker social support, with psychosocial safety climate (PSC) as an upstream predictor and moderator of this relationship. Using hierarchical linear modelling (*N* = 380 frontline healthcare workers, nested within *N* = 63 teams) on longitudinal data, we found a reciprocal relationship between burnout and co-worker support, which was both triggered and moderated by PSC. These findings provide initial evidence for a social support–burnout loss spiral, wherein individuals with poor co-worker support are more likely to become burnt-out, and in this depleted state they are subsequently less likely to reach out for those social supports when available, which would, in turn, lead to further burnout. This social support–burnout loss spiral is exacerbated when working in a low-PSC context, as the environment does not send positive safety signals about resource scarcity and replenishment. Therefore, PSC may be a potential target for intervention both in primary prevention (i.e., stopping the loss spiral in the first instance) and in secondary intervention, as high PSC signals to workers already in the burnout–support loss cycle when it is safe to reinvest resources or engage in recovery behavior.

## 1. Introduction

Burnout is often characterized by “emotional exhaustion”, which has emerged as a key safety outcome in the area of workplace wellbeing [1]. Whilst research has identified workplace factors that lead to burnout, little is known about how burnout develops over time. A better understanding of how burnout is perpetuated is crucial for informing the development of effective intervention efforts, especially given the flow-on effects that burnout has for accidents and incidents in high-risk industries. Interventions may either break cycles of loss or prevent them from developing in the first place, in order to improve workplace wellbeing and promote primary prevention.

### 1.1. Background

To begin to understand how burnout is perpetuated over time, we turn first to the conservation of resources (COR) theory [2,3]. Hobfoll posits that individuals seek to acquire and protect valued resources, so stress occurs when these resources are threatened, lost, or unattainable in spite of resource investment [2,3,4]. In general, resource loss spirals can occur when that initial resource loss leads to further losses in a negative downward spiral, and there is building evidence for such loss spirals in the workplace setting, impacting on wellbeing (e.g., [5,6,7]). These spirals are maintained by the confounding effect of associated stress, the greater impact of resource loss on wellbeing than resource gain, and a reduction in the resources available to mitigate further losses [4]. As such, loss spirals are thought to increase in magnitude and momentum with each cycle, thereby having a deleterious effect on wellbeing [4].

In the context of burnout, the existing body of research has focused predominantly on loss spirals between burnout and other individual (i.e., intraindividual) resources, such as presenteeism [8] and job insecurity [9]. However, Halbesleben and Wheeler [6] highlighted the need for research examining the role of interpersonal (i.e., interindividual) resources such as social support, which may help us to better understand how burnout is perpetuated in the workplace. This knowledge is valuable beyond our extant understanding of loss spirals in general, because interpersonal resources play a critical role in the interconnected nature of the modern workplace.

### 1.2. The Reciprocal Relationship between Burnout and Co-Worker Social Support

Workplace social support (for example, co-worker support) has been described as the degree to which a worker perceives that their wellbeing is valued by others in their workplace, as demonstrated by positive interpersonal interactions and provision of resources [10]. A major advantage of investigating a loss outcome that is interpersonal in nature is that it provides a clear target for intervention over which workers have a degree of control (e.g., fostering supportive interactions between colleagues on a daily basis). Whilst it is true that social support has already been established as having the potential for intervention according to existing knowledge [11], reframing further exploration of the relationship between burnout and social support as one that is reciprocal in nature offers us the opportunity to better understand how burnout changes over time, therefore better equipping us to prevent burnout. By looking deeper into the reciprocal relationship between burnout and social supports, we may better understand the longer-lasting impacts of this potential loss spiral and, therefore, how interventions that operate on interindividual resources could intervene in downward cycles of loss (playing a role in the primary and secondary prevention of accidents). Thus, for the present study, co-worker social support was selected as a variable of interest, in order to investigate how burnout may be perpetuated over time.

In generating our hypotheses, we began by first recognizing that co-worker social support has been identified as playing a key role in the etiology of the various dimensions of burnout [12]. Social support in general has a demonstrated protective effect against workplace stressors [13] and makes it less likely for workers to be exposed to social demands such bullying and harassment.

Therefore, we predict the following:

**H1:** 
*Co-worker social support will be a significant negative predictor of burnout at a subsequent timepoint.*


In tandem, burnout is expected to undermine co-worker support itself by reducing the emotional capacity of workers to invest in and maintain social networks in the workplace. Furthermore, workers who are in an emotionally depleted state may be less likely to offer support to others and, therefore, unlikely to build a reciprocal social relationship. As a result, workers are unable to draw on co-worker support to reduce the likelihood of strain and to mitigate the impact of stressors, further exacerbating their levels of burnout. The meta-analysis by Halbesleben [12] established a clear negative correlation between the “emotional exhaustion” dimension of burnout and workplace social support.

Therefore, we predict the following:

**H2:** 
*Burnout will be a significant negative predictor of co-worker social support at a subsequent timepoint.*


With H1 and H2, we began to address Halbesleben’s [12] call for longitudinal research to examine a possible reciprocal relationship between burnout and social support.

### 1.3. Psychosocial Safety Climate as an Upstream Predictor

Not all factors influencing burnout are within the control of co-workers, especially within an organization that has extant issues around workplace safety culture. Individuals may even try to support one another and take responsibility for hostile working conditions that are actually the result of a broader systemic disregard for worker health in favor of productivity and organizational outcomes. More than a decade of research evidence points to psychosocial safety climate (PSC) as a lead indicator of psychological health outcomes via job demands and resources [14], and to this day PSC research provides a strong framework for understanding the origins of health in the workplace (e.g., see the systematic review by Golzad [15]). PSC refers to shared perceptions of how much an organization values the psychological health and safety of their workers, as evidenced by policies, practices, and procedures [16]. The level of PSC within an organization influences workplace conditions such as job demands and resources; in high-PSC workplaces, the psychological wellbeing of workers is valued, resulting in increased resources (such as co-worker social support) and work engagement via a motivational pathway [14]. In low-PSC workplaces, the psychological health of workers is not considered to be a priority, leading to increased job demands and psychological health problems (such as burnout) via a health erosion pathway [14]. Therefore, PSC poses a potential target for intervention to prevent the development of the reciprocal negative relationship between burnout and poor co-worker support, but this has yet to be tested.

Prior research has linked PSC and burnout through work pressure and emotional demands [16]. Low PSC is likely to trigger the burnout–support loss spiral by creating an environment wherein workers have increased demands and limited resources to protect against these demands, resulting in greater vulnerability to burnout. Workers may then have a limited ability to alter the situation and, therefore, continue to experience demands or lack of resources, along with the lack of recovery of resources, in turn leading to further resource loss.

For PSC, we begin by predicting the following:

**H3:** 
*Psychosocial safety climate (PSC) will be a significant negative predictor of burnout at a subsequent timepoint.*


More specifically, we expect that this relationship will exist at both the individual level (H3a), and the team level (H3b). Multilevel approaches are important for climate measures such as PSC, as they are often conceptualized at a ”group level” rather than just the workers’ individual experiences [17]. Group perceptions of shared constructs like organizational climate provide a valuable—and in some cases more reliable—perspective on how PSC may impact on burnout, over and above individual perceptions. But how might PSC operate within the loss spiral itself?

### 1.4. The Moderating Role of Psychosocial Safety Climate

Based on the body of evidence, PSC may also act as a moderator by activating (in high-PSC environments) or blocking (in low-PSC environments) access to resources required to intervene in the loss spiral. As outlined by COR theory, workplace conditions are thought to act as gatekeepers to resource caravan passageways [4]. In essence, resources do not exist in isolation; they tend to aggregate and exist within a social environment, and that dynamic context is critical to the maintenance erosion of those resources [18]. We argue that PSC is an important organizational safety context that could inform COR’s resource caravan passageway when explaining loss cycles. For example, when PSC is low, the loss spiral may be further perpetuated by an inability to access the resources required to mitigate the impact of resource loss. In contrast, when PSC is high, the loss spiral may be broken by workers having access to the necessary support to rebuild lost resources and to protect against future losses. High PSC might also act as a safety signal encouraging workers to fully utilize other resources at their disposal (cf. [14]). In high-PSC environments, resources are more plentiful; according to COR theory, this means that resource loss will not be as salient as compared with environments wherein resources are scarcer. This suggests that PSC may play some kind of moderating role in loss spirals; however, this has yet to be fully explored, e.g., the evidence so far only suggests that PSC moderates the *unidirectional* relationship between emotional demands and emotional resources [16]. We therefore have an opportunity to apply a COR lens to the role that PSC plays in loss cycles, because PSC may send safety signals to workers about resource scarcity and replenishment. Could the PSC context suggest when it is safe to reinvest resources or engage in recovery behavior and, thus, offset a downward spiral of loss?

Therefore, we aim to extend COR theory by linking it with the PSC theory using the extended JDR model via a multilevel method to examine whether PSC acts as (1) a trigger for burnout at both the team and individual levels (see H3), and as (2) a moderator for the resource loss spiral between burnout and co-worker support.

**H4:** 
*Psychosocial safety climate (PSC) will moderate the reciprocal relationship between burnout and co-worker social support.*


## 2. Materials and Methods

In the present study, we selected frontline healthcare workers as our target population because of the well-demonstrated links between PSC and worker safety (e.g., see [19]) and patient safety [20], as well as because the nursing workplace hierarchy is suitable for planning multilevel models. We used a longitudinal multilevel design to explore the impacts of both the individual and collective experiences of PSC (i.e., team perceptions) on workers’ resource loss erosion. That is, group perceptions of PSC (i.e., group level) and individual perceptions of PSC (i.e., individual level) can both significantly explain the levels of demand and resources experienced in the workplace, as well as workers’ feelings of burnout [17]. Furthermore, by aggregating PSC to the work unit rather than relying solely upon individual perceptions, we were able to gain a more accurate estimate of PSC levels and their impact upon the resource loss spiral within the team [21]. Lastly, the multilevel approach controlled for the shared experiences that frontline healthcare workers are exposed to within their teams when exploring the individual-level pathways from emotional exhaustion to co-worker support. We were interested in how both team and individual perceptions of PSC may trigger the burnout–support loss spiral, using a longitudinal multilevel approach.

### 2.1. Participants and Procedure

Data were matched according to the team unit in which the individual worked, which resulted in 63 teams and 380 employees, who participated in all three waves of the study. The participants were drawn from three hospitals across a variety of healthcare occupational groups, including medical, nursing, allied health, admin/corporate, and aides. The researchers were physically present at each site and approached potential participants during their shifts, ensuring that all employees had fair opportunity to participate should they wish. The researchers visited the same teams across all possible shifts (early, late, and night) over the course of one to three weeks per team (according to staff pool size), and they checked the shift rosters to verify broad coverage of the staff pool. The majority of the participants involved in the study were female (81%), who were aged 40–49 (27%) and 50–59 (30%). These participants completed a self-report survey on an iPad during their work shift, disseminated by three research assistants. Data collection occurred in half-yearly blocks, with Time 1 collected in the second half of 2015, Time 2 in the first half of 2016, and Time 3 in the second half of 2016.

### 2.2. Measures

#### 2.2.1. Psychosocial Safety Climate

Extant safety culture is typically measured via climate scales; if the pervasive social context in the workplace is akin to a weather pattern, then a PSC scale acts like a thermometer to take a snapshot of the climate (i.e., the prevailing temperature, in this analogy). Therefore, we measured PSC with the established 12-item psychosocial safety climate scale (PSC-12 [22]), which consists of four subscales. The scale was adapted slightly to measure team PSC in contrast to the previously measured organizational PSC [22], so as to better reflect our two-level HLM approach. The specific change involved adding the clarifier “In my team…” to the beginning of each statement, but the actual items themselves remained identical to the established PSC-12 scale [22]. The four subscales—management commitment, organizational communication, management priority, and organizational participation—comprised 3 items each. Example items include “Management shows support for stress prevention through involvement and commitment” (management commitment), “There is good communication here about psychological issues which affect me” (organizational communication), “Management acts decisively when a concern of an employee’s psychological status is raised” (management priority), and “Employees are encouraged to become involved in psychological health and safety matters” (organizational participation). Each item was measured using a 5-point Likert scale, with scores ranging from 1 (strongly disagree) to 5 (strongly agree). High scores on the scale indicated high levels of team PSC.

#### 2.2.2. Burnout

Burnout was measured using the 5-item emotional exhaustion subscale from the Maslach Burnout Inventory [23]. Emotional exhaustion was selected because Halbesleben [12] found the relationship between social support and burnout to be stronger than other dimensions, such as depersonalization and personal accomplishment. The participants rated their levels of emotional exhaustion using a 7-point Likert scale ranging from 1 (not at all) to 7 (always). An example item from this scale is “I feel emotionally drained from my work”. High scores on the scale indicate high levels of emotional exhaustion at work.

#### 2.2.3. Co-Worker Support

Co-worker support was measured using the 3-item subscale from the JCQ 2.0 developed by the Job Content Questionnaire Centre [24]. This subscale consists of items such as “People I work with are friendly”. These items were measured on a 4-point Likert scale ranging from 1 (strongly disagree) to 4 (strongly agree). High scores on this scale reflect high levels of co-worker support.

### 2.3. Analysis

To test the pathways proposed in Figure 1, we used Hierarchical Linear Modelling (HLM) v7 software and Process v2.16 as an add-on in the Statistical Package for the Social Sciences (SPSS) v22. Typical quantitative research in psychology explores relationships between variables that are experienced by people (i.e., the “individual level”), but HLM is a statistical approach that allows researchers to also cluster those people according to a shared aspect of their environment (e.g., children in different classes, or different schools), so as to simultaneously test whether those relationships between variables exist at a higher level, i.e., involved in more complex group-level processes [25]. This makes HLM well positioned for organizational research because it allows us to explore relationships between variables from the perspective of an individual worker, as well as the clustered experiences of their team, and even nested within higher-order groupings (such as divisions, worksites, or franchises) when such data are available. In this study, we used a two-level model with the individual workers (*N* = 380) nested within teams (*N* = 63) that had a clear identifiable leader to whom all members reported. We used SPSS to create the two data files for use in HLM. The Level 1 data file was at the individual level. The Level 2 data file was aggregated to the team level. Data were matched between the two files based on their team.

To test PSC Time 1 at the team level (Level 2) with respect to workers’ emotional exhaustion, we aggregated the individual level scores by team. One-way ANOVAs showed significant between-team variance for PSC (*F*(62, 336) = 2.28, *p* < 0.001) and emotional exhaustion (*F*(62, 362) = 1.50, *p* < 0.05). The ICC scores for PSC Time 1 and emotional exhaustion Time 2 were 0.17 (*χ*^2^ (62) = 117.27, *p* < 0.001) and 0.08 (*χ*^2^ (62) = 93.93, *p* < 0.01), respectively. This justifies a multilevel analysis strategy.

In longitudinal research, it is standard practice to control for previous timepoints of the dependent variables. However, in testing the hypotheses for the present study, we did not have sufficient power to control for previous timepoints due to the expectation of small effect sizes for this sample. Specifically, to find a significant relationship, a sample size of 476 would be needed to detect the small effect sizes that were expected for the association from PSC to emotional exhaustion, from emotional exhaustion to co-worker support, and from co-worker support to emotional exhaustion [26].

To test the proposed PSC extended resource loss model (see Figure 1), we first simultaneously assessed the paths from team-level and individual-level PSC at Time 1 onto emotional exhaustion at Time 2, consistent with the work of [17], who argued that researchers need to simultaneously test both individual- and group-level measures of PSC. Next, we assessed the first stage of the loss spiral involving the pathway leading from emotional exhaustion at Time 2 to the outcome, co-worker support, at Time 3. Finally, in three steps, we tested the second stage of the loss spiral involving the PSC (Time 2)-moderated reverse pathway going from co-worker support at Time 2 to emotional exhaustion at Time 3. In the first step, we tested the direct path from co-worker support to emotional exhaustion. In the second step, while controlling for the direct pathway from co-worker support to emotional exhaustion, we tested both PSC and the interaction term between PSC and co-worker support simultaneously. To create the interaction term for PSC and co-worker support, we multiplied the two mean-centered products of individual-level PSC and co-worker support.

The final step was to test the range of significance for the moderation of PSC (Time 2) on the relationships between co-worker support (Time 2) and emotional exhaustion (Time 3). We used the Johnson–Neyman technique in Process (Hayes, 2013). The Johnson–Neyman technique identifies at what levels of the moderator (PSC) the conditional effect of the independent variable (i.e., co-worker support) on the dependent variable (i.e., emotional exhaustion) is statistically significant, along with the strength of the moderation effect. Within the figures plotting the range of significance obtained from the Johnson–Neyman technique, a dotted vertical line was used to indicate the level of the moderator at which the moderation transitions from non-significant to significant. Finally, the pick-a-point approach was used to visually plot the moderation.

## 3. Results

### 3.1. Descriptive Statistics

Means, standard deviations, and correlations between the variables of interest are presented in Table 1. The correlation analysis, as expected, revealed a significant positive correlation between PSC at Time 1 and PSC at Time 2. Similarly, significant positive correlations were found between emotional exhaustion at Time 2 and emotional exhaustion at Time 3, and between co-worker support at Time 2 and Time 3.

At the individual level, significant negative correlations were found between PSC (Time 1 and Time 2) and emotional exhaustion at both timepoints, along with significant positive correlations with co-worker support at both timepoints. The analysis also revealed that the correlations between emotional exhaustion and co-worker support were negative and significant.

Age was significantly associated with emotional exhaustion (Time 2 and Time 3), in that participants who were older had lower levels of emotional exhaustion. Hours worked in the last week across the three timepoints were all negatively correlated with emotional exhaustion at Time 3, while emotional exhaustion at Time 2 was only correlated with hours worked at Time 1 and Time 2, as expected. As such, working more hours during the week was associated with higher levels of emotional exhaustion. In addition, PSC at Time 1 was positively correlated with hours worked at Time 2. As such, individuals who reported that their teams had higher levels of PSC spent more hours in the workplace. Gender was not significantly correlated with any of the variables of interest except for hours worked, indicating that males worked higher numbers of hours during the week at Time 2 and Time 3, but there were no differences between males and females in their levels of PSC, emotional exhaustion, or co-worker support.

### 3.2. Hypothesis Testing

As part of the first and second stages of the resource loss spiral, we hypothesized that (H1) co-worker social support at Time 2 would have a significant negative relationship with emotional exhaustion at the following timepoint (Time 3), and likewise, the loss spiral could be seen in the opposite direction (H2), wherein emotional exhaustion at Time 2 would have a significant negative relationship with co-worker support at the following timepoint (Time 3). Support for the resource loss spiral can be seen in Figure 2, which displays the results of the HLM analyses; both Hypotheses 1 and 2 were supported, with the specific details of the moderation of the H1 pathway discussed further below.

Once support for the H1 and H2 resource loss spiral was established, we tested the role of PSC as a leading indicator (H3) and moderator (H4). As expected, the HLM results demonstrated that Time 1 PSC at both the team level (Level 2) and the individual level (Level 1) was significantly related to emotional exhaustion at Time 2, in the anticipated direction (see Figure 2). That is, PSC at both levels—team and individual—simultaneously predicted emotional exhaustion 5 months later, suggesting that PSC plays a leading indicator role in predicting the start of a loss cycle, which supports Hypotheses H3a and H3b.

H4 tested the moderating role of PSC at Time 2 on the relationship between co-worker support (Time 2) and emotional exhaustion (Time 3). In the first step, we confirmed the existence of the direct effect of co-worker support on emotional exhaustion (H1), and we found a significant negative pathway (*γ* = −0.53 (0.24), *p* < 0.05). In the second step, we added both the moderator (PSC) and the interaction term, in addition to co-worker social support. With the addition of PSC and the interaction term, the direct pathway from co-worker support to emotional exhaustion was no longer significant (*γ* = −0.46 (0.25), *p* = 0.07). However, both PSC (*γ* = −0.08 [0.03], *p* < 0.05), and the moderation interaction term were significant (see Figure 2), supporting the moderating role of PSC in the resource loss cycle that was predicted in Hypothesis 4, the extent of which is explored next.

### 3.3. Moderation Analysis

The Johnson–Neyman technique of moderation in the SPSS macro Process v2.16.3 [27] was used to check the range of significance for the PSC-moderated pathway from co-worker support to burnout. While controlling for the main effects of both co-worker support and PSC, the interaction between co-worker support and PSC was significant. Next, using the Johnson–Neyman technique of moderation, the conditional effects of co-worker support (Time 2) on emotional exhaustion (Time 3) were plotted as a function of PSC (i.e., point estimate), along with the 95% ULCI and LLCI (see Figure 3). We were able to identify the level of the moderator at which the moderation transitions from significant to non-significant using α = 0.05 (plotted using a vertical dotted line; see Figure 3). It was found that when the PSC levels in the team were above a score of 39.54, the conditional impact of co-worker support on emotional exhaustion decreased as levels of PSC increased.

To visualize the moderation, the pick-a-point approach was used (see Figure 4). In conditions of low co-worker support, high levels of PSC were useful in reducing the levels of emotional exhaustion experienced by the workers. Meanwhile, when co-worker support and PSC were both low, emotional exhaustion levels were at their highest. As expected when working in a low-PSC context, the climate exacerbates the resource loss cycle.

## 4. Discussion

All four hypotheses were supported, suggesting the possible existence of a resource loss cycle between co-worker social support and emotional exhaustion (H1 and H2) that is both initiated (H3) and moderated (H4) by PSC. In essence, a person working in an environment with poor protections for psychosocial safety is more likely to experience burnout, which then compromises their ability to invest in and maintain social resources; this lack of co-worker support leads again to burnout, forming a reciprocal downward spiral that is exacerbated even further in low-PSC contexts.

Our discovery is important in the context of the literature, because there is limited understanding of how the climate of an organization influences potential loss spirals, as well as of whether and how such cycles of loss can be interceded to improve the wellbeing of workers. For example, Hobfoll [4] highlighted the need for further research investigating workplace conditions as gatekeepers to resource caravan passageways (i.e., pathways to groups of resources). Our results demonstrate that both team and individual perceptions of PSC triggered a burnout–loss cycle, potentially addressing the question “under what conditions can a resource loss cycle commence?” Interestingly, whilst we found that low co-worker social support increased feelings of burnout (H1), this pathway in the loss cycle only existed as a function of PSC (moderation H4). As the PSC levels increased, the association between co-worker support and burnout decreased, which is consistent with the proposition that high levels of PSC would help workers escape from the burnout–support loss spiral.

However, whilst we found that PSC at both levels predicted the start of a loss cycle, it was only PSC at the individual level that moderated that reciprocal relationship between co-worker support and burnout. We were unable to establish PSC as a team-level moderator of the loss cycle, which may be due to a relatively simple statistical limitation of power (since moderation is easier to detect at the individual level) or a more complex interaction, such as the existence of a distal effect when positive and negative constructs cluster differently (e.g., see MacKinnon [28], who posits that it is difficult to detect interactions between opposite variables). Nevertheless, at the individual level, it might be construed that when individual perceptions of PSC were low, increasing the levels of co-worker support did little to improve workers’ feelings of burnout. Even if the team’s PSC at the group level is high, the signals from the environment are at odds with the interpersonal experience, which may lead to workers lacking the confidence to invest personal resources in co-workers to offset the disparity. It might be the case that it takes resource investment to capitalize on social support (for example, knowing what and how to share or delegate to others, or taking time off work to debrief), so without high PSC sending clear safety signals, co-worker social support itself is simply insufficient.

### 4.1. Theoretical Implications

The current study provides preliminary evidence for a burnout–support loss spiral (see Figure 5). Building on prior unidirectional research linking burnout and social support [12], our results showed a reciprocal relationship between emotional exhaustion and co-worker support. This unifies established theoretical knowledge from multiple frameworks in new ways, namely, how co-worker social support leads to emotional exhaustion, as proposed in the extended JDR model [14], and how emotional exhaustion erodes social supports as proposed by the COR model [12]. The synthesis of these two trains of thought into a reciprocal relationship provides a richer understanding of how burnout and social support are intertwined, emphasizing the importance of considering interindividual factors in relation to the maintenance of burnout.

Our findings further build upon COR theory by elucidating the processes through which loss spirals may be broken. The results identified PSC as a potential moderator of the burnout–support loss spiral. Applying the principles of COR theory [4], PSC may break the loss spiral by creating a workplace environment that facilitates access to resource caravan passageways that can be used to mitigate resource loss. Interestingly, when PSC was low, increased co-worker support had little impact on burnout. This finding suggests that the benefits of co-worker support may only be activated in workplace contexts where PSC is high, because high-PSC contexts may act as a safety signal indicating that it is safe for workers to reach out for support from others in the team to help cope with stressors [14,29]. This represents a fusion of both PSC’s and COR’s theoretical perspectives; whereas PSC frameworks suggest that the climate is a leading indicator, under a COR lens the ecological conditions influence the degree of access to resources [4], and PSC can also operate as that environmental signal. This is critical in healthcare contexts, where entrenched cultural norms may encourage workers to endure through psychosocial hazards, be “resilient”, and focus on individual coping strategies in the face of systemic job design and organizational issues [30]. However, in a high-PSC workplace, accessing social support from co-workers may be encouraged and seen as a positive, resulting in recovery from stressors and reinforcement of this behavior [29]. Therefore, high PSC may be a precondition for accessing the benefits of resources such as the co-worker support required to break from a loss spiral. Another possible mechanism through which PSC may influence the burnout–support loss spiral is psychological need thwarting (i.e., obstruction of psychological needs resulting in feelings of oppression, inadequacy, or rejection [31]), with a recent study identifying psychological need thwarting as a mediator in the relationship between PSC and burnout [32]. This may explain why PSC acts as an upstream predictor of the burnout–support loss spiral in our results, but it may also be due to low levels of PSC increasing job demands as well as blocking resource acquisition to offset those demands. Overall, our study offers new perspectives on how multiple established theoretical frameworks can be applied together in order to better understand the novel relationships of interindividual resources with burnout, burnout–support loss cycles, and the multiple roles that PSC plays in preventing and breaking these cycles.

### 4.2. Practical Implications

Our findings suggest that there is an opportunity for a new perspective on designing interventions. It is common nomenclature in industry to label safety interventions as primary, secondary, or tertiary, with a sliding scale from prevention through to treatment of symptomatic outcomes. Whilst the former is certainly key to avoiding significant costs to the worker as well as to the organization, little is known about the effectiveness of interventions once a loss spiral has already been triggered.

We argue that PSC presents a twofold opportunity for intervention in the burnout–support relationship. First, improving PSC plays a traditional preventative role [30] wherein, for example, psychosocial awareness workshops at the management level or organization-wide systems like customized OHS websites integrating PSC [33] represent an upstream factor for job design and resourcing, which would protect workers from psychosocial hazards. However, our evidence suggests that, simultaneously, at the lower level, improving PSC might be a useful intervention point to arbitrate a loss spiral that is already in progress. An intervention at the frontline level might include PSC training workshops for teams [30], collaborating with organizational citizenship building [34], or even using safety apps and wearables [35] and real-time monitoring dashboards for PSC [30], any of which can assist with the recognition, management, and reporting of psychosocial risks; these should now incorporate the role of co-worker support. Workers often inherently understand the importance of social networks, debriefing, and lending a “helping hand” when a co-worker is in need, but a psychosocial intervention might better explain the relationships between these factors (in intercepting a fellow worker caught in the tide of a loss spiral). Important new research shows that PSC interventions are effective when using a participatory approach, with a needs assessment (from surveys and during workshops) guiding the development of a tailored action plan [36]. We would advocate that PSC trainers now teach the important relationship between PSC and the support–burnout loss cycle. PSC interventions might also serve to normalize the open discussion of psychosocial factors and send a clear safety signal, because workers might only take advantage of social support in high-PSC contexts. This approach could not only assist affected individuals but also prompt co-workers capable of providing support to do so. We propose that our findings are useful not only in designing workshop content for bespoke PSC interventions, but also in other initiatives like workplace partnership strategies [37], wherein workers and unions have high involvement at all levels of decision-making, as well as strategies for bolstering co-worker support such as enhancing employees’ voice [38].

### 4.3. Limitations and Future Research

First, instrumentation presents a key limitation. Although co-worker social support is an established JCQ2.0 scale, participants are still rating their co-workers’ provision of social support, which may only capture actions that are noticed rather than objective data on attempts to support fellow team members. A burnt-out worker may not recognize when resources are present and may underrate levels of co-worker social support. Our data are also susceptible to common-method bias (due to the use of self-report surveys), which may inflate the observed relationships. However, we addressed this bias to some extent through the temporal separation of variables, and by including team aggregated PSC. The latter provides a more objective measure of the levels of PSC in the team than relying solely on individual perceptions. However, due to sample size limitations, we did not control for the stability of the dependent variables with previous levels of the variable. Despite the limited ability to make causal claims for the pathways from burnout to social support, and from social support to burnout, prior longitudinal research has demonstrated that PSC predicts future burnout. For example, Dollard and Bakker [16] found a longitudinal association between PSC and burnout, with increasing levels of PSC at Time 1 leading to decreasing levels of burnout at Time 3, controlling for Time 1 burnout.

Our sample of frontline healthcare workers consisted primarily of female workers between the ages of 30 and 59, on a full-time contract. The ability to generalize our findings to work groups with different demographics may be limited, but these statistics do reflect the demographics of the nursing profession at least. Future studies could address generalizability issues by investigating the PSC extended burnout–support loss spiral in samples from different industries, such transportation, construction, education, policing, academia, and service industries, as outlined in a recent systematic review of PSC research [39], or even across general workers. Further, the PSC extended burnout–support loss spiral must be explored in different cultural contexts because the body of recent research points to noticeable differences in PSC around the world, especially between Western and Eastern countries [40], which may be due to disparities in union density and bargaining power for better working conditions. There are also well-known gender differences in supportive co-worker relations (e.g., see [41]), so we suggest that future research should explore whether gender plays a role in the PSC extended burnout–support loss spiral.

In terms of theory, our study calls attention to three potential directions for future research. First, research is needed to examine interventions targeting the burnout–loss spiral. For example, researchers could develop and test a primary intervention (i.e., a PSC training workshop) for the prevention of the burnout–support loss spiral, as well as an intervention that arbitrates the loss spiral. Second, research could expand on the current study to incorporate investigation into the impact of the burnout–support loss spiral on key organizational outcomes such as patient safety (e.g., medication errors) and staff safety incidents (e.g., needle-stick injuries). Lastly, it is possible that burnout may have a reciprocal relationship with other interindividual factors, so future research could further answer Halbesleben and Wheeler’s [6] call to explore the bigger picture in interindividual resource cycles, beyond just co-worker social support.

## 5. Conclusions

Our study provides initial evidence for a burnout–support loss cycle, wherein burnt-out individuals have reduced capacity to foster social networks in the workplace, resulting in limited access to supportive peers and, in turn, greater vulnerability to burnout. In addition, PSC was identified as a trigger and moderator of this downward spiral of loss. With the potential for prevention *and* the ability to break an ongoing burnout–support loss spiral, our study suggests that PSC is a promising target for new intervention strategies to improve workplace wellbeing.

## Figures and Tables

**Figure 1 healthcare-11-03168-f001:**
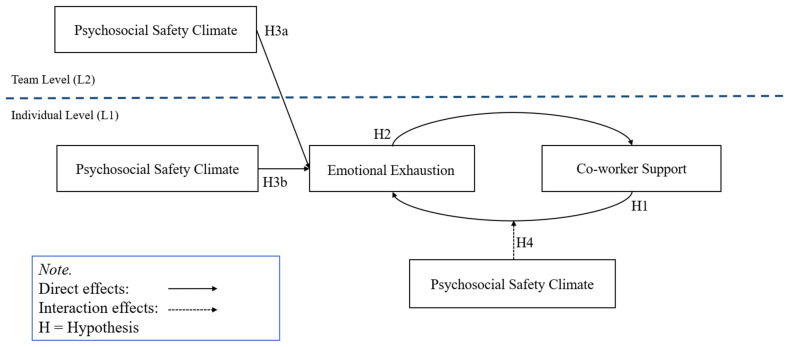
Path diagram showing the proposed theoretical model. L = Level.

**Figure 2 healthcare-11-03168-f002:**
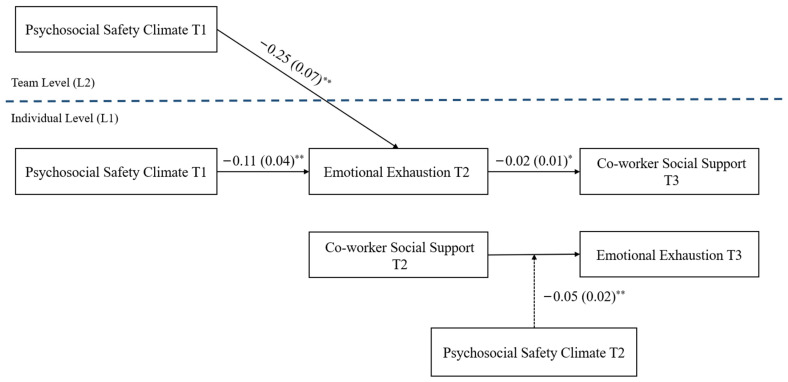
The psychosocial safety climate (PSC)–moderated burnout (i.e., emotional exhaustion)–social support (i.e., co-worker support) loss spiral. T = time; L = level; * = *p* < 0.05; ** *p* < 0.01.

**Figure 3 healthcare-11-03168-f003:**
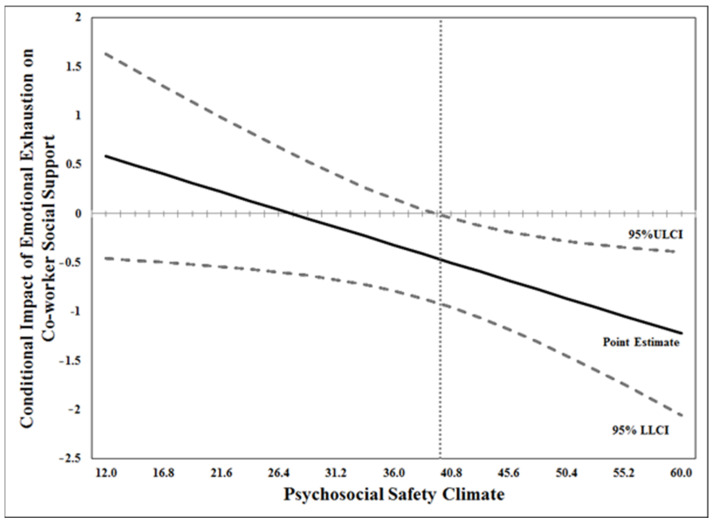
The conditional effect of co-worker support on emotional exhaustion as a function of psychosocial safety climate.

**Figure 4 healthcare-11-03168-f004:**
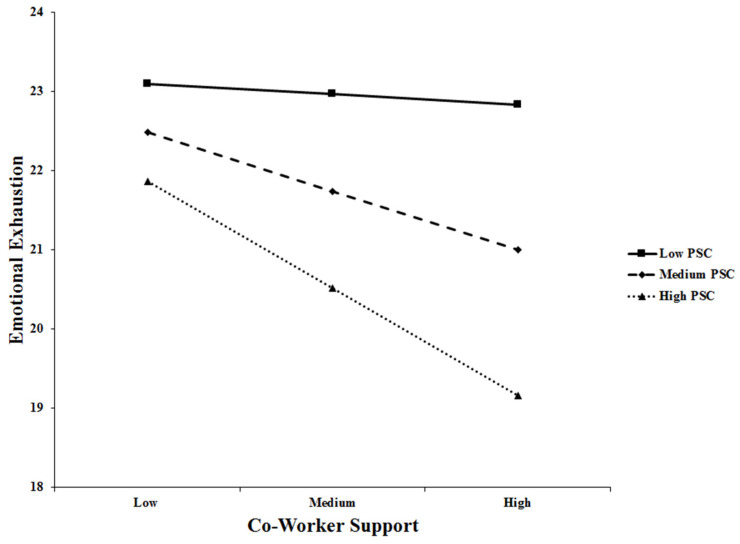
The moderation of the effect of co-worker support on emotional exhaustion by psychosocial safety climate. PSC = psychosocial safety climate.

**Figure 5 healthcare-11-03168-f005:**
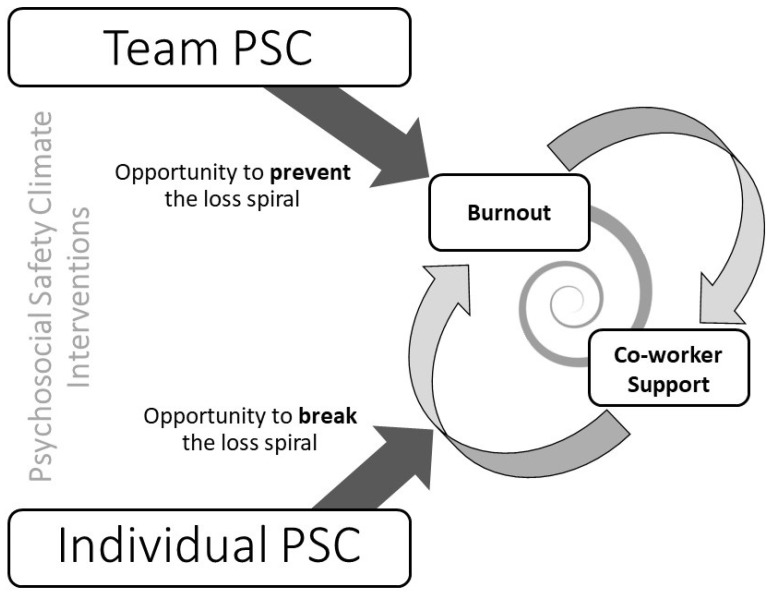
An illustration of the PSC extended burnout–support loss spiral.

**Table 1 healthcare-11-03168-t001:** Descriptive statistics.

	M	SD	1.	2.	3.	4.	5.	6.	7.	8.	9.	10.
1. Age	3.94	1.17	-									
2. Gender	1.15	0.36	0.07	-								
3. Hours Worked T1	4.01	0.93	−0.03	0.10 ^†^	-							
4. Hours Worked T2	3.98	0.93	−0.08	0.10 *	0.46 ***	-						
5. Hours Worked T3	4.05	0.86	0.04	0.13 **	0.39 ***	0.44 ***	-					
6. PSC T1	39.28	10.40	−0.01	0.01	0.05	0.11 *	0.03	-				
7. PSC T2	40.72	10.88	−0.03	0.07	0.07	0.03	−0.04	0.58 ***	-			
8. Emotional Exhaustion T2	21.75	7.09	−0.18 ***	−0.04	0.12 *	0.11 *	0.09 ^†^	−0.20 ***	−0.34 ***	-		
9. Emotional Exhaustion T3	21.51	6.74	−0.18 ***	−0.01	0.12 *	0.13 **	0.16 **	−0.18 **	−0.22 ***	0.54 ***	-	
10. Co-worker Support T2	9.87	1.48	−0.07	−0.04	0.03	−0.01	−0.08 ^†^	0.19 ***	0.39 ***	−0.20 ***	−0.16 **	-
11. Co-worker Support T3	9.83	1.54	−0.02	−0.07	0.05	−0.01	−0.09 ^†^	0.15 ***	0.27 **	−0.15 **	−0.15 **	0.45 ***

Note. Gender = lower value female, higher value male; PSC = psychosocial safety climate; Emotional Exhaustion = burnout; T = time; *** *p* < 0.001, ** *p* < 0.01, * *p* < 0.05, ^†^ *p* < 0.10.

## Data Availability

Data are contained within the article; Further data (not publicly available due to ethics) are available on request from the corresponding author.

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
