# Peer review of "Evidence for a Negative Loss Spiral between Co-Worker Social Support and Burnout: Can Psychosocial Safety Climate Break the Cycle?"

_healthcare, 2023, doi:10.3390/healthcare11243168_

Round 1
Reviewer 1 Report
Comments and Suggestions for Authors
I have completed my review of your manuscript, which investigates the reciprocal relationship between burnout and co-worker social support, with a particular focus on the role of Psychosocial Safety Climate (PSC) as both a predictor and a moderator. The study is well-conceived and addresses a significant gap in the current literature on workplace wellbeing, particularly in healthcare settings. Below are my suggestions for minor revisions that could further strengthen the manuscript:
1. **Clarification of Methodological Details:**
- While your methodology is robust, providing additional details about the hierarchical linear modeling approach and the rationale for choosing this method could enhance the reader's understanding. Specifically, a brief explanation of why this method is suitable for your data structure (N=380 frontline health care workers, nested within n=63 teams) would be beneficial.
2. **Expansion on PSC Conceptualization:**
- The manuscript would benefit from a more detailed discussion of the conceptualization of PSC, particularly how it is operationalized in your study. This would help readers unfamiliar with the concept to better understand its role and significance in your findings.
3. **Discussion of Limitations:**
- While you acknowledge the limitations related to common method bias and sample characteristics, a more thorough discussion regarding how these limitations might impact the generalizability of your findings would be valuable. Additionally, consider discussing any limitations in the measurement tools used for assessing burnout and co-worker support.
4. **Implications for Future Research:**
- Your suggestions for future research are insightful. However, expanding on how future studies might explore the applicability of your findings in different cultural or organizational contexts could be beneficial. This would add depth to the understanding of the universality of the burnout-support loss spiral.
5. **Practical Implications:**
- The practical implications section is compelling, but it could be enhanced by providing more specific examples or case studies of how organizations can implement interventions targeting PSC. This would offer practical value to readers looking to apply your findings in real-world settings.
6. **Graphical Representations:**
- Consider including graphical representations or models to illustrate the reciprocal relationship between burnout and co-worker support, and how PSC interacts with these variables. Visual aids can enhance comprehension and engagement with the material.
7. **Reference Updates:**
- Ensure that all references are up-to-date and consider including any recent studies that might have been published after your literature review was conducted. This would ensure that your manuscript reflects the most current state of research in this area.
Best regards,
Author Response
We thank the reviewer for their high praise; we are glad to hear about the recognition of the importance of our study. The suggested revisions are excellent and we have taken the opportunity to strengthen our manuscript in light of feedback. Please find below our itemised responses to your comments:
Reviewer 1, Comment #1: Provide details on what hierarchical linear modeling is, with a brief explanation of the rationale for choosing this approach.
As per recommendation, in order to better enhance reader understanding of the methods used we have included a brief explanation of HLM (drawing on a useful citation from Woltman et al., 2012), and related this to the structure of the data in our specific study. See the new highlighted section on Page 5-6.
Reviewer 1, Comment #2: Provide more detail on the conceptualization of PSC, particularly how it is operationalized in your study.
We have now included a passage (see new highlighted section on Page 3) to help readers unfamiliar with PSC, and then a specific reference to how we operationalize PSC in organizational research later on Page 5 (lines 206 through 210).
Reviewer 1, Comment #3a: Discuss generalizability limitations
We address both this and Comment #4 in a combined new section on Page 13.
Reviewer 1, Comment #3b: Discuss limitation in measuring burnout or social support
This is an important observation, especially given that participants in a burnt-out state may not recognize when resources are available, and therefore underrate co-worker social support in self-report scales. We have now included a discussion of these limitations on Page 13.
Reviewer 1, Comment #4: Future research in different cultural / organizational contexts.
We have now included more specific future directions that include cultural contexts and different industries as targets (see highlighted segment on Page 13).
Reviewer 1, Comment #5: Enhance the practical implications section with more specific examples of interventions targeting PSC.
We have now drawn on recent publications that demonstrate different ways of implementing PSC interventions (four new passages on Page 12) which as per your suggestion will help provide practical value to readers looking to apply the findings.
Reviewer 1, Comment #6: Consider including graphical representations or models to illustrate the reciprocal relationship between burnout and co-worker support, and how PSC interacts with these variables.
Although we had included Figure 1 to explain the model, we recognize the potential for a new visual aid specifically illustrating the loss spiral and points of intervention could enhance comprehension and engagement. We propose to include this in the Theoretical Implications (see Page 11) because it is the best point at which to explain how the new model interacts with existing theory. The full size figure is available in attachments.
Reviewer 1, Comment #7: Include new references in the area.
Thank you for pointing this out, we realise that in the years we’ve been writing this paper we had fully integrated some of the latest research. We have now addressed this, including half a dozen trending new studies that have since been published (2022 and 2023) which you will find highlighted in the reference list on pages 14 and 15.
Reviewer 2 Report
Comments and Suggestions for Authors
The paper offers a better understanding on burnout. It provides the novel idea that PSC may function as a moderator by activating or blocking access to resources required to intervene in the loss spiral.
The investigation of the factor of co-worker support is also a brilliant idea.
The discovery is that a person working in an environment with poor protection for psychosocial safety is more likely to experience burnout, which might affect their ability to invest in and maintain social resources. This lack of co-worker support leads to burnout.
The manuscript provides initial evidence for the burnout support loss cycle, where burnout individuals have reduced capacity to foster social networking at the workplace. This results in limited access to supportive peers which in turn leads to greater vulnerability to burnout.
The study provides serious theoretical implications, i.e. a richer understanding on how burnout and social support are intertwined, by emphasizing on the importance of considering inter-individual factors in relation to the maintenance of burnout.
In method, authors should clarify how they chose participants in order to assure the representativeness of the population. Furthermore, in unit 2.2.1. could they elaborate on how they adopted the PSC scales (line 199)?
Regarding the practical implications, the findings could be useful not only in planning tailor-made workshops but also with other initiatives, such as workplace partnership, that could empower a psychosocial context against burnout.
In addition, employee voice can practically foster co-worker support and to enhance social partners’ capacity building within enterprises.
Regarding their proposals for further research authors could investigate the role of gender in co-worker support.
Lastly, there is a lack of references written in the last 5 years. Authors could update the references.
Author Response
We thank the reviewer for their high praise; we are glad to hear about the recognition of the importance of our study. The suggested revisions are excellent and we have taken the opportunity to strengthen our manuscript in light of feedback. Please find below our itemised responses to your comments:
Reviewer 2, Comment #1a: Clarify participant selection
More detail is now provided in 2.1 re participant selection, opportunities to participate, and representativeness (see Page 4).
Reviewer 2, Comment #1b: Elaborate on how the PSC scales were adapted for the study
We have now properly clarified the modification, namely the statement “In my team…” which was added to the beginning of each PSC item to clearly refer to their team, rather than the PSC of their whole organization (Page 5, lines 212-215).
Reviewer 2, Comment #2: Consider additional practical implications
We have incorporated these excellent suggestions in our practical implications. Please refer to all the new content on Page 12, but specifically the mention of workplace partnership and employee voice (lines 497-501)
Reviewer 2, Comment #3: Consider future research on the role of gender in co-worker support
We have a variety of new considerations in the future research directions, including cultural context, gender, and different industrial settings (see Page 13).
Reviewer 2, Comment #4: Lack of references written in the last 5 years
Thank you for pointing this out, we realise that in the years we’ve been writing this paper we had fully integrated some of the latest research. We have now addressed this, including half a dozen trending new studies that have since been published (2022 and 2023) which you will find highlighted in the reference list on pages 14 and 15.